# Randomized Simplicial Hessian Update

**Árpád Bűrmen \***, **Tadej Tuma** and **Jernej Olenšek**

Faculty of Electrical Engineering, University of Ljubljana, Tržaška Cesta 25, SI-1000 Ljubljana, Slovenia; tadej.tuma@fe.uni-lj.si (T.T.); jernej.olensek@fe.uni-lj.si (J.O.)
\* Correspondence: arpad.buermen@fe.uni-lj.si; Tel.: +386-1-4768-322

**Abstract:** Recently, a derivative-free optimization algorithm was proposed that utilizes a minimum Frobenius norm (MFN) Hessian update for estimating the second derivative information, which in turn is used for accelerating the search. The proposed update formula relies only on computed function values and is a closed-form expression for a special case of a more general approach first published by Powell. This paper analyzes the convergence of the proposed update formula under the assumption that the points from $\mathbb{R}^n$ where the function value is known are random. The analysis assumes that the $N + 2$ points used by the update formula are obtained by adding $N + 1$ vectors to a central point. The vectors are obtained by transforming a prototype set of $N + 1$ vectors with a random orthogonal matrix from the Haar measure. The prototype set must positively span a $N \le n$ dimensional subspace. Because the update is random by nature we can estimate a lower bound on the expected improvement of the approximate Hessian. This lower bound was derived for a special case of the proposed update by Leventhal and Lewis. We generalize their result and show that the amount of improvement greatly depends on $N$ as well as the choice of the vectors in the prototype set. The obtained result is then used for analyzing the performance of the update based on various commonly used prototype sets. One of the results obtained by this analysis states that a regular $n$-simplex is a bad choice for a prototype set because it does not guarantee any improvement of the approximate Hessian.

**Keywords:** derivative-free optimization; Hessian update; random matrices; uniform distribution

**MSC:** 90C56; 90C53; 65K05; 15A52

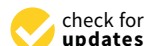

## 1. Introduction

Derivative-free optimization algorithms have attracted much attention due to the fact that in many optimization problems, the evaluation of the gradients of the function subject to optimization and constraints is expensive. Such optimization problems can be often formulated as constrained black-box optimization (BBO) [1] problems of the form

$$\min f(\mathbf{x}) \quad \text{subject to} \tag{1}$$

$$c_i(\mathbf{x}) \le 0 \quad i = 1, 2, \ldots, n_C \tag{2}$$

Functions $f$ and $c_i$ are maps from $\mathbb{R}^n$ to $\mathbb{R}$. The objective is to minimize $f$ subject to $n_C$ nonlinear constraints defined by functions $c_i$. The method for computing $f$ and $c_i$ is treated as a black-box, and the gradients are usually not available. Such problems often arise in engineering optimization when simulation is used for obtaining the function values. BBO often relies on models of the function and of the constraints. Various approaches to building black-box models were developed in the past, such as linear [2] and quadratic models [3], radial-basis functions [4], support vector machines [5], neural networks [6], etc.

In this paper, we focus on the quadratic models of $f$ and $c_i$. The most challenging task in building these models is the computation of the Hessian matrix. Instead of using the exact Hessian, the model can utilize an approximate Hessian. The approximation can be

improved gradually by applying an update formula based on the function and the gradient values at points visited in the algorithm's past. As the algorithm converges towards a solution, the approximate Hessian converges to the true Hessian.

For derivative-based optimization, several approaches for updating the approximate Hessian are well studied and tested in practice (e.g., BFGS update, SR1 update [7]). Unfortunately, these approaches rely on the gradient of the function (constraints), which, by assumption, is not available in derivative-free optimization.

Let $n$ denote the dimension of the search space. For derivative-free optimization, a Hessian update formula based on the function values computed at $m \geq n + 2$ points visited in the algorithm's past was proposed by Powell in [8]. The update formula was obtained by minimizing the Frobenius norm of the update applied to the approximate Hessian subject to linear constraints imposed by the function values at $m$ points in the search space. The paper proposed an efficient way for computing the update and explored some of its properties. The convergence rate of the update formula was not studied.

In a later paper, a simple update formula that uses three collinear points for computing the updated approximate Hessian [9] was examined. The normalized direction along which the three points lie was assumed to be uniformly distributed on the unit sphere. With this assumption, the convergence rate of the update was analyzed and shown to be linear. This update formula was successfully used in a derivative-free algorithm from the family of mesh adaptive direct search algorithms (MADS) [10]. A similar Hessian updating approach was used for speeding up global optimization in [11].

The assumption that the points taking part in an update must be collinear is a significant limitation for the underlying derivative-free algorithm. With this in mind, a new simplicial update formula was proposed in [12]. The formula relies on $m \leq n + 2$ points. The reason for choosing the term simplicial Hessian update is the fact that the $m - 1$ points form a simplex centered around the first point. For $m = n + 2$, the formula is a special case of the update formula proposed in [8]. By imposing some restrictions on the positions of the $m$ points, the update formula can be used for any $m$ that satisfies $3 \leq m \leq n + 2$. The case $m = 3$ corresponds to the update formula proposed in [9].

To illustrate the approach for obtaining the update formula, let us assume that the current quadratic model of function $f$ is given by

$$m(\mathbf{x}) = \frac{1}{2}\mathbf{x}^\mathsf{T}\mathbf{B}\mathbf{x} + \hat{\mathbf{g}}^\mathsf{T}\mathbf{x} + \hat{c} \tag{3}$$

$\mathbf{B}$ is the current approximate Hessian. Let the points where the function is known be denoted by $\mathbf{x}_i$. For the sake of simplicity, let $f_i$ denote $f(\mathbf{x}_i)$. Based on these points, we are looking for an updated model:

$$m_+(\mathbf{x}) = \frac{1}{2}\mathbf{x}^\mathsf{T}\mathbf{B}_+\mathbf{x} + \hat{\mathbf{g}}_+^\mathsf{T}\mathbf{x} + \hat{c}_+. \tag{4}$$

The model must satisfy $m$ constraints

$$m_+(\mathbf{x}_i) = f_i \tag{5}$$

that are linear in $\hat{c}$ and the components of $\mathbf{B}_+$ and $\hat{\mathbf{g}}_+$. Based on these constraints, we are looking for an updated approximate Hessian $\mathbf{B}_+$. Because we have fewer constraints than there are unknowns, we also require that $\|\mathbf{B}_+ - \mathbf{B}\|_\mathrm{F}$ is minimal ($\|\cdot\|_\mathrm{F}$ denotes the Frobenius norm). The update formula we obtain in this way is a minimum Frobenius norm update formula.

For computing the expected improvement of the approximate Hessian, we first assume $f$ itself is quadratic. We also assume the aforementioned $m$ points are obtained by applying a random orthogonal transformation to $m - 1$ vectors that form a prototype set and adding the resulting vectors to a central point. As in [9], the convergence rate of the update is linear.

The speed with which the approximate Hessian converges to the true Hessian depends on the choice of the prototype set. Our result is a generalization of the result published in [9].

This paper is divided as follows. In Section 2, some basic properties of minimum Frobenius norm updates are explored. The Frobenius product is revisited with the purpose of simplifying the notation, and the update formula is derived. In the next section, uniformly distributed orthogonal matrices are introduced. Some auxiliary results are derived that are later used for computing the expected improvement of the approximate Hessian. Section 4 analyzes the convergence of the proposed update and derives the expected value of the improvement in the sense of the Frobenius norm of the difference between the approximate Hessian and the true Hessian. The expected improvement is computed for several prototype sets. The section is followed by an example demonstrating the convergence of the proposed update and concluding remarks.

**Notation.** Components of vectors (**a**) and matrices (**A**) are denoted by subscripts (i.e., $a_i$ and $a_{ij}$, respectively). The $i$-th column of matrix **A** is denoted by $\mathbf{a}_i$. The unit vectors forming an orthogonal basis for $\mathbb{R}^n$ are denoted by $\mathbf{e}_i$. Vectors are assumed to be column vectors, and the inner product of two vectors is written in matrix notation as $\mathbf{a}^\mathrm{T}\mathbf{b}$. The Frobenius norm and the trace of a matrix are denoted by $\|\cdot\|_\mathrm{F}$ and $\mathrm{tr}(\cdot)$, respectively. The expected value of a random variable is denoted by $\mathrm{E}[\cdot]$.

## 2. Obtaining the Update Formula

Let **H** denote the Hessian of a function. Minimum Frobenius norm (MFN) update formulas replace the current Hessian approximation **B** with a new (better) approximation $\mathbf{B}_+$ in such manner that the Frobenius norm of the change (i.e., $\mathbf{B}_+ - \mathbf{B}$) is minimal, subject to constraints imposed on $\mathbf{B}_+$.

The Frobenius norm is a norm induced by the Frobenius (inner) product on the space of $n$-by-$n$ matrices. The Frobenius product of two matrices is given by

$$\mathbf{A} : \mathbf{B} = \sum_{i=1}^{n}\sum_{j=1}^{n} a_{ij}b_{ij} = \mathrm{tr}(\mathbf{A}^\mathrm{T}\mathbf{B}) = \mathrm{tr}(\mathbf{B}^\mathrm{T}\mathbf{A}). \tag{6}$$

Using the Frobenius product, one can express the Frobenius norm of matrix **A** as

$$\|\mathbf{A}\|_\mathrm{F}^2 = \mathbf{A} : \mathbf{A}. \tag{7}$$

Quadratic terms can be expressed with the Frobenius product as

$$\mathbf{x}^\mathrm{T}\mathbf{A}\mathbf{x} = \mathbf{A} : (\mathbf{x}\mathbf{x}^\mathrm{T}). \tag{8}$$

The Frobenius product introduces the notion of perpendicularity into the set of matrices (not to be confused with the orthogonality of matrices, which is equivalent to $\mathbf{Q}^\mathrm{T}\mathbf{Q} = \mathbf{I}$).

**Definition 1.** *Two nonzero matrices **A** and **B** are perpendicular (denoted by **A** $\perp$ **B**) if **A** : **B** = 0.*

The Frobenius product can also be used for expressing linear constraints. A linear equality constraint on matrix **X** can be formulated as

$$\mathbf{A} : \mathbf{X} = a. \tag{9}$$

The following Lemma provides motivation for the use of minimum Frobenius norm updating.

**Lemma 1.** *Let **H**, **B**, and $\mathbf{B}_+$ denote the exact, the current approximate, and the updated approximate Hessian, respectively. Suppose we have $m$ linear equality constraints of the form*

$$\mathbf{A}_i : \mathbf{B}_+ = a_i, \quad i = 1,\ldots,m. \tag{10}$$

imposed on $\mathbf{B}_+$. Let $\mathcal{P}_\perp$ denote the subspace spanned by matrices $\mathbf{A}_i$. Then, the corresponding MFN update satisfies

1. $(\mathbf{B}_+ - \mathbf{B}) \in \mathcal{P}_\perp$, and
2. $\|\mathbf{B}_+ - \mathbf{H}\|_F \le \|\mathbf{B} - \mathbf{H}\|_F$.

**Proof.** Finding the MFN update is equivalent to minimizing the Frobenius norm of $\mathbf{B}_+ - \mathbf{B}$ subject to linear equality constraints (10). These constraints define an affine subspace in the $n(n+1)/2$ dimensional space of Hessian matrices, and $\mathbf{B}_+$ is a member of this affine subspace. Because the true Hessian also satisfies constraints (10), it is also a member of the aforementioned affine subspace.

To simplify the problem, we can translate it in such manner that $\mathbf{H}$ becomes $\mathbf{0}$. When we do this, the linear constraints become homogeneous, and instead of an affine subspace, they now define an ordinary subspace $\mathcal{P}$. Its orthogonal complement $\mathcal{P}_\perp$ is spanned by matrices $\mathbf{A}_i$. Due to translation, $\mathbf{B}$ and $\mathbf{B}_+$ are replaced by $\mathbf{B} - \mathbf{H}$ and $\mathbf{B}_+ - \mathbf{H}$, of which the latter is a member of $\mathcal{P}$. Points with constant $\|\mathbf{B}_+ - \mathbf{B}\|_F = \|\mathbf{B}_+ - \mathbf{H} - (\mathbf{B} - \mathbf{H})\|_F$ lie on a sphere centered at $\mathbf{B} - \mathbf{H}$. Matrix $\mathbf{B}_+ - \mathbf{H}$ that corresponds to the smallest $\|\mathbf{B}_+ - \mathbf{B}\|_F$ lies on a sphere centered at $\mathbf{B} - \mathbf{H}$ that is tangential to subspace $\mathcal{P}$. Therefore, $\mathbf{B}_+ - \mathbf{B}$ must be perpendicular to $\mathcal{P}$, i.e., $\mathbf{B}_+ - \mathbf{B} \in \mathcal{P}_\perp$. This proves the first claim.

Due to $\mathbf{B}_+ - \mathbf{H} \in \mathcal{P}$, we can see that $\mathbf{B}_+ - \mathbf{H}$ and $\mathbf{B}_+ - \mathbf{B}$ are perpendicular. From $\mathbf{B} - \mathbf{H} = \mathbf{B}_+ - \mathbf{H} - (\mathbf{B}_+ - \mathbf{B})$, we have

$$\|\mathbf{B} - \mathbf{H}\|_F^2 = \|\mathbf{B}_+ - \mathbf{H}\|_F^2 + \|\mathbf{B}_+ - \mathbf{B}\|_F^2 \tag{11}$$

The second claim immediately follows from this result. □

Consider a quadratic function

$$q(\mathbf{x}) = \frac{1}{2}\mathbf{x}^T\mathbf{H}\mathbf{x} + \mathbf{g}^T\mathbf{x} + c \tag{12}$$

where $\mathbf{H}$ is its Hessian and $\mathbf{g}$ its gradient at $\mathbf{x} = \mathbf{0}$. Let the current and the updated approximation to $q(\mathbf{x})$ be given by

$$m(\mathbf{x}) = \frac{1}{2}\mathbf{x}^T\mathbf{B}\mathbf{x} + \hat{\mathbf{g}}^T\mathbf{x} + \hat{c} \tag{13}$$

and

$$m_+(\mathbf{x}) = \frac{1}{2}\mathbf{x}^T\mathbf{B}_+\mathbf{x} + \hat{\mathbf{g}}_+^T\mathbf{x} + \hat{c}_+, \tag{14}$$

respectively. In MFN, updating $\mathbf{B}_+$ is obtained by minimizing $\|\mathbf{B}_+ - \mathbf{B}\|_F$. The following lemma introduces one such update based on the case when the value of $q$ is known at $N + 2$ points.

**Lemma 2.** *Let $q_0, \dots, q_{N+1}$, where $N \le n$ denote the values of $q(\mathbf{x})$ corresponding to distinct points $\mathbf{x}_0, \dots, \mathbf{x}_{N+1}$, respectively. Let $\mathbf{v}_i = \mathbf{x}_i - \mathbf{x}_0$ and assume $\sum_{i=1}^{N+1} \alpha_i \mathbf{v}_i = \mathbf{0}$ with at least one $\alpha_i \ne 0$. Then the simplicial MFN update satisfying the interpolation conditions $m_+(\mathbf{x}_i) = q_i$ for $i = 0, 1, \dots, N+1$ can be computed as*

$$\mathbf{B}_+ = \mathbf{B} + \beta\mathbf{A} \tag{15}$$

*where*

$$\mathbf{A} = \frac{1}{2}\sum_{i=1}^{N+1} \alpha_i \mathbf{v}_i\mathbf{v}_i^T, \tag{16}$$

$$\beta = \frac{\sum_{i=1}^{N+1}\alpha_i\left(\mathbf{v}_i^T(\mathbf{H} - \mathbf{B})\mathbf{v}_i\right)}{2\|\mathbf{A}\|_F^2} = \frac{\sum_{i=1}^{N+1}\alpha_i\left(2(q_i - q_0) - \mathbf{v}_i^T\mathbf{B}\mathbf{v}_i\right)}{2\|\mathbf{A}\|_F^2}. \tag{17}$$

**Proof.** By assumption we have

$$
\begin{aligned}
q_i = q(\mathbf{x}_i) &= \frac{1}{2}\mathbf{x}_i^{\mathrm{T}}\mathbf{H}\mathbf{x}_i + \mathbf{g}^{\mathrm{T}}\mathbf{x}_i + c \\
&= \frac{1}{2}(\mathbf{x}_0 + \mathbf{v}_i)^{\mathrm{T}}\mathbf{H}(\mathbf{x}_0 + \mathbf{v}_i) + \mathbf{g}^{\mathrm{T}}(\mathbf{x}_0 + \mathbf{v}_i) + c
\end{aligned}
\tag{18}
$$

Due to the interpolation conditions, we have $N + 2$ constraints

$$
q_i = m_+(\mathbf{x}_i) = \frac{1}{2}(\mathbf{x}_0 + \mathbf{v}_i)^{\mathrm{T}}\mathbf{B}_+(\mathbf{x}_0 + \mathbf{v}_i) + \hat{\mathbf{g}}_+^{\mathrm{T}}(\mathbf{x}_0 + \mathbf{v}_i) + \hat{c}_+
\tag{19}
$$

By subtraction, we eliminate $\hat{c}_+$ and obtain $N + 1$ constraints

$$
q_i - q_0 = \frac{1}{2}\mathbf{v}_i^{\mathrm{T}}\mathbf{B}_+\mathbf{v}_i + (\hat{\mathbf{g}}_+ + \mathbf{B}_+\mathbf{x}_0)^{\mathrm{T}}\mathbf{v}_i \quad i = 1, .., N+1.
\tag{20}
$$

Multiplying (20) with $\alpha_i$ and adding the resulting equations yields

$$
\frac{1}{2}\sum_{i=1}^{N+1}\alpha_i\mathbf{v}_i^{\mathrm{T}}\mathbf{B}_+\mathbf{v}_i + (\hat{\mathbf{g}}_+ + \mathbf{B}_+\mathbf{x}_0)^{\mathrm{T}}\sum_{i=1}^{N+1}\alpha_i\mathbf{v}_i = \sum_{i=1}^{N+1}\alpha_i(q_i - q_0).
\tag{21}
$$

By assumption, the second term on the left-hand side of (21) vanishes (thus, $\hat{\mathbf{g}}_+$ is eliminated). We are left with a single linear constraint on $\mathbf{B}_+$:

$$
\frac{1}{2}\sum_{i=1}^{N+1}\alpha_i\mathbf{v}_i^{\mathrm{T}}\mathbf{B}_+\mathbf{v}_i = \sum_{i=1}^{N+1}\alpha_i(q_i - q_0)
\tag{22}
$$

which can be rewritten by recalling (8) as

$$
\mathbf{A} : \mathbf{B}_+ = \sum_{i=1}^{N+1}\alpha_i(q_i - q_0),
\tag{23}
$$

where

$$
\mathbf{A} = \frac{1}{2}\sum_{i=1}^{N+1}\alpha_i\mathbf{v}_i\mathbf{v}_i^{\mathrm{T}}.
\tag{24}
$$

Equation (23) is a linear constraint on the updated Hessian approximation $\mathbf{B}_+$. This is the only constraint on $\mathbf{B}_+$. From Lemma 1, we can see that $\mathcal{P}_\perp$ is spanned by $\mathbf{A}$. Therefore, we can write

$$
\mathbf{B}_+ - \mathbf{B} = \beta\mathbf{A}.
\tag{25}
$$

By computing the Frobenius product of (25) with $\mathbf{A}$ and taking into account (23), we arrive at

$$
\sum_{i=1}^{N+1}\alpha_i(q_i - q_0) - \mathbf{A} : \mathbf{B} = \beta\mathbf{A} : \mathbf{A}.
\tag{26}
$$

Now we can compute $\beta$:

$$
\beta = \frac{\sum_{i=1}^{N+1}\alpha_i(q_i - q_0) - \mathbf{A} : \mathbf{B}}{\mathbf{A} : \mathbf{A}} = \frac{\sum_{i=1}^{N+1}\alpha_i\left(2(q_i - q_0) - \mathbf{v}_i^{\mathrm{T}}\mathbf{B}\mathbf{v}_i\right)}{2\|\mathbf{A}\|_{\mathrm{F}}^2}.
$$

□

The simplicial update formula introduced by Lemma 2 is the closed-form solution of the equations arising from the MFN update in [8] for $N = n$. One can see this by comparing the interpolation conditions to those in [8]. Due to the assumption $\sum_{i=1}^{N+1}\alpha_i\mathbf{v}_i = \mathbf{0}$, we can also apply it when $N < n$. The assumption implies the points $\mathbf{x}_1, \ldots, \mathbf{x}_{N+1}$ are positioned

in a specific manner with respect to $\mathbf{x}_0$ (i.e., there exists a nontrivial linear combination $\sum_{i=1}^{N+1} \alpha_i (\mathbf{x}_i - \mathbf{x}_0) = \mathbf{0}$).

By choosing $N = 1$, we obtain a special case of the simplicial MFN update, where all three distinct points must be collinear to satisfy $\sum_{i=1}^{N+1} \alpha_i \mathbf{v}_i = \mathbf{0}$. Suppose $\mathbf{v}_1 = -\mathbf{v}_2 = \mathbf{v}$ and $\alpha_1 = \alpha_2 = 1$. Then,

$$\mathbf{A} = \mathbf{v}\mathbf{v}^{\mathrm{T}}, \tag{27}$$

$$\beta = \frac{(q_1 + q_2 - 2q_0) - \mathbf{v}^{\mathrm{T}}\mathbf{B}\mathbf{v}}{\|\mathbf{v}\|^4} = \frac{q_{\mathbf{v}}^{(2)}(\mathbf{x}_0) - \mathbf{v}^{\mathrm{T}}\mathbf{B}\mathbf{v}}{\|\mathbf{v}\|^4} \tag{28}$$

where $q_{\mathbf{v}}^{(2)}(\mathbf{x}_0) = \mathbf{v}^{\mathrm{T}}\mathbf{H}\mathbf{v}$ is the second directional derivative of $q$ along direction $\mathbf{v}$. The convergence properties of this MFN update formula were analyzed in [9]. The formula was used in the derivative-free optimization algorithm proposed in [10].

## 3. Uniformly Distributed Orthogonal Matrices

The notion of a uniform distribution over the group of orthogonal matrices ($\mathcal{O}_n$) can be introduced via the Haar measure [13]. Let $\mathbf{A}$ denote a matrix with independent normally distributed elements with zero mean and variance 1. A random orthogonal matrix from the Haar measure ($\mathbf{O}$) can then be obtained with Algorithm 1.

---

**Algorithm 1** Constructing a random orthogonal matrix from the Haar measure.

---

1. Perform QR decomposition $\mathbf{A} = \mathbf{Q}\mathbf{R}$.
2. Construct a diagonal matrix $\mathbf{D}$ with $d_{ii} = 1$ if $r_{ii} \geq 0$ and $d_{ii} = -1$ otherwise.
3. $\mathbf{O} = \mathbf{Q}\mathbf{D}$.

---

Multiplying $\mathbf{O}$ with any unit vector results in a random unit vector that is uniformly distributed on the unit sphere ($\mathcal{S}_n$) [14]. It can be shown that $\mathbf{O}^{\mathrm{T}}$ is also a uniformly distributed orthogonal matrix. Consequently, every column and every row of $\mathbf{O}$ are a random unit vector with uniform distribution on $\mathcal{S}_n$.

The results of this section are obtained with the help of the following lemma.

**Lemma 3.** *Let $x \in \mathbb{R}^n$ and let $d\sigma$ denote the surface element of $\mathcal{S}_n$. Then,*

$$V_n(r_1, r_2, \ldots, r_n) = \int_{\|\mathbf{x}\|=1} x_1^{2r_1} x_2^{2r_2} \cdot \cdots \cdot x_n^{2r_n} d\sigma = \frac{2 \prod_{i=1}^{n} \Gamma(r_i + 1/2)}{\Gamma(n/2 + \sum_{i=1}^{n} r_i)} \tag{29}$$

**Proof.** See [15], Appendix B. $\square$

From Lemma 3, we can obtain the surface area of $\mathcal{S}_n$ by choosing $r_1 = \ldots = r_n = 0$.

$$S_n = \frac{2\Gamma(1/2)^n}{\Gamma(n/2)} \tag{30}$$

Let $\mathbf{o}_i$ and $\mathbf{o}_j$ denote two random vectors that correspond to the $i$-th and the $j$-th column of $\mathbf{O}$. If $i \neq j$, then $\mathbf{o}_i^{\mathrm{T}}\mathbf{o}_j = 0$. We denote the $k$-th component of $\mathbf{o}_i$ as $o_{ki}$.

**Lemma 4.** *Let* **O** *be a uniformly distributed random orthogonal matrix. Then,*

$$\mathrm{E}\left[\left(\mathbf{e}_k^{\mathrm{T}}\mathbf{o}_i\right)^2\right] = \mathrm{E}\left[o_{ki}^2\right] = \frac{1}{n}, \tag{31}$$

$$\mathrm{E}\left[\left(\mathbf{e}_k^{\mathrm{T}}\mathbf{o}_i\right)^2\left(\mathbf{e}_l^{\mathrm{T}}\mathbf{o}_i\right)^2\right] = \mathrm{E}\left[o_{ki}^2 o_{li}^2\right] = \begin{cases} \frac{3}{n(n+2)} & k = l \\ \frac{1}{n(n+2)} & k \neq l \end{cases}, \tag{32}$$

$$\mathrm{E}\left[\left(\mathbf{e}_k^{\mathrm{T}}\mathbf{o}_i\right)^2\left(\mathbf{e}_l^{\mathrm{T}}\mathbf{o}_j\right)^2\right] = \mathrm{E}\left[o_{ki}^2 o_{lj}^2\right] = \begin{cases} \frac{1}{n(n+2)} & k = l, \ i \neq j \\ \frac{n+1}{(n-1)n(n+2)} & k \neq l, \ i \neq j \end{cases}. \tag{33}$$

**Proof.** For proving (31), we can assume without loss of generality $k = i = 1$. Because $\mathbf{o}_1 = \mathbf{u}$ is uniformly distributed on $\mathcal{S}_n$, the expected value of $f(\mathbf{u})$ can be obtained by computing the mean value of $f(\mathbf{u})$ over $\mathcal{S}_n$. We use Lemma 3 for expressing the integral over the surface of $\mathcal{S}_n$.

$$\mathrm{E}\left[o_{ki}^2\right] = \mathrm{E}\left[o_{11}^2\right] = \mathrm{E}\left[u_1^2\right] = S_n^{-1}\int_{\|\mathbf{u}\|=1}u_1^2\mathrm{d}\sigma \tag{34}$$

$$= S_n^{-1}\frac{2\Gamma(3/2)\Gamma(1/2)^{n-1}}{\Gamma(1+n/2)} = \frac{1}{n}. \tag{35}$$

Regarding (32) for $k = l$, we have

$$\mathrm{E}\left[o_{ki}^2 o_{li}^2\right] = \mathrm{E}\left[o_{ki}^4\right] = S_n^{-1}\int_{\|\mathbf{u}\|=1}u_1^4\mathrm{d}\sigma \tag{36}$$

$$= S_n^{-1}\frac{2\Gamma(5/2)\Gamma(1/2)^{n-1}}{\Gamma(2+n/2)} = \frac{3}{n(n+2)}. \tag{37}$$

For $k \neq l$, we assume without loss of generality $k = 1$, $l = 2$. From Lemma 3, we have

$$\mathrm{E}\left[o_{ki}^2 o_{li}^2\right] = S_n^{-1}\int_{\|\mathbf{u}\|=1}u_1^2 u_2^2\mathrm{d}\sigma \tag{38}$$

$$= S_n^{-1}\frac{2\Gamma(3/2)^2\Gamma(1/2)^{n-2}}{\Gamma(2+n/2)} = \frac{1}{n(n+2)} \tag{39}$$

For (33) with $k = l$, we can show that it is identical to (32) with $k \neq l$. We have

$$\mathbf{e}_k^{\mathrm{T}}\mathbf{o}_i = \mathbf{e}_k^{\mathrm{T}}\mathbf{O}\mathbf{e}_i = \mathbf{e}_i^{\mathrm{T}}\mathbf{O}^{\mathrm{T}}\mathbf{e}_k = \mathbf{e}_i^{\mathrm{T}}\left(\mathbf{O}^{\mathrm{T}}\right)_k \tag{40}$$

where $\left(\mathbf{O}^{\mathrm{T}}\right)_k$ is the $k$-th column of $\mathbf{O}^{\mathrm{T}}$. This implies

$$\left(\mathbf{e}_k^{\mathrm{T}}\mathbf{o}_i\right)^2\left(\mathbf{e}_k^{\mathrm{T}}\mathbf{o}_j\right)^2 = \left(\mathbf{e}_i^{\mathrm{T}}\left(\mathbf{O}^{\mathrm{T}}\right)_k\right)^2\left(\mathbf{e}_j^{\mathrm{T}}\left(\mathbf{O}^{\mathrm{T}}\right)_k\right)^2 \tag{41}$$

To confirm (33) for $k = l$, we take into account that $\mathbf{O}^{\mathrm{T}}$ is also a random orthogonal matrix from the Haar measure, replace $\mathbf{O}$ with $\mathbf{O}^{\mathrm{T}}$ in (32), and rename $i$, $k$, and $l$ to $k$, $i$, and $j$, respectively.

Finally, to prove (33) for $k \neq l$, we can assume without loss of generality $i = k = 1$ and $j = l = 2$. The cosine of the angle between $\mathbf{o}_1$ and $\mathbf{e}_2$ can be expressed as $o_{21} = \mathbf{e}_2^{\mathrm{T}}\mathbf{o}_1 = \cos\phi$. Random vector $\mathbf{o}_2$ is orthogonal to $\mathbf{o}_1$. Its realizations cover a unit sphere in an $n-1$-dimensional subspace $\mathcal{B}$ orthogonal to $\mathbf{o}_1$. Unit vectors $\mathbf{b}_1, \ldots, \mathbf{b}_{n-1}$ form an orthogonal basis for this subspace. Note that $\mathbf{b}_i^{\mathrm{T}}\mathbf{o}_1 = 0$. The conditional probability density distribution of $\mathbf{o}_2$ is uniform on the aforementioned unit sphere in $\mathcal{B}$. Vector $\mathbf{o}_2$ can be expressed as

$$\mathbf{o}_2 = \sum_{i=1}^{n-1}\eta_i\mathbf{b}_i, \qquad \sum_{i=1}^{n-1}\eta_i^2 = 1, \tag{42}$$

where vector $[\eta_1, \ldots, \eta_{n-1}]$ is uniformly distributed on $\mathcal{S}_{n-1}$. Without loss of generality we can choose vectors $\mathbf{b}_i$ in such manner that $\mathbf{e}_2 = \mathbf{o}_1 \cos \phi + \mathbf{b}_1 \sin \phi$, where $\phi$ is the angle between $\mathbf{e}_2$ and $\mathbf{o}_1$. Now we have

$$o_{22} = \mathbf{e}_2^{\mathsf{T}} \mathbf{o}_2 = \sum_{i=1}^{n-1} (\mathbf{o}_1 \cos \phi + \mathbf{b}_1 \sin \phi)^{\mathsf{T}} \eta_i \mathbf{b}_i = \eta_1 \sin \phi \tag{43}$$

and

$$o_{11}^2 o_{22}^2 = (\mathbf{e}_1^{\mathsf{T}} \mathbf{o}_1)^2 (\mathbf{e}_2^{\mathsf{T}} \mathbf{o}_2)^2 = o_{11}^2 \eta_1^2 \sin^2 \phi = \eta_1^2 o_{11}^2 \left(1 - o_{21}^2\right). \tag{44}$$

Next, we can express

$$\mathrm{E}\left[o_{ki}^2 o_{lj}^2\right] = \mathrm{E}\left[\eta_1^2\right] \mathrm{E}\left[o_{11}^2 - o_{11}^2 o_{21}^2\right], \tag{45}$$

where the first expected value refers to $[\eta_1, \ldots, \eta_{n-1}] \in \mathcal{S}_{n-1}$ and the second one to $\mathbf{o}_1 \in \mathcal{S}_n$. Using Lemma 3 and previously proven (32), we arrive at

$$\mathrm{E}\left[\eta_1^2\right] \mathrm{E}\left[o_{11}^2 - o_{11}^2 o_{21}^2\right] = \frac{1}{n-1} \left(\frac{1}{n} - \frac{1}{n(n+2)}\right) = \frac{n+1}{(n-1)n(n+2)} \tag{46}$$

□

## 4. Convergence of the Proposed Update

Multiplying vectors in a prototype set $\mathcal{D} = \{\mathbf{d}_1, \ldots, \mathbf{d}_{N+1}\}$ with a uniformly distributed random orthogonal matrix $\mathbf{O}$ results in a set of random vectors $\mathcal{V} = \{\mathbf{v}_1, \ldots, \mathbf{v}_{N+1}\}$ such that every $\mathbf{v}/\|\mathbf{v}\|$ is uniformly distributed on $\mathcal{S}_n$. The angles between the vectors in a realization of such a set are identical to the angles between the corresponding vectors from the prototype set.

Suppose one is interested in the expected amount of improvement resulting from one application of the update formula from Lemma 2. We assume that the $N + 2$ points where the function value is computed comprise $\mathbf{x}_0$ and additional $N + 1$ points generated using a random orthogonal matrix $\mathbf{O}$ and a prototype set of vectors $\{\mathbf{d}_1, \ldots, \mathbf{d}_{N+1}\}$ in the following manner.

$$\mathbf{x}_i = \mathbf{x}_0 + \mathbf{O}\mathbf{d}_i = \mathbf{x}_0 + \mathbf{v}_i, \qquad i = 1, \ldots, N+1. \tag{47}$$

First, we prove an auxiliary lemma.

**Lemma 5.** *Let* $\mathbf{a}$, $\mathbf{b}$, *and* $\mathbf{O}$ *denote two unit vectors with* $\cos \varphi = \mathbf{a}^{\mathsf{T}} \mathbf{b}$ *and a uniformly distributed orthogonal matrix, respectively. Let* $\mathbf{u} = \mathbf{O}\mathbf{a}$ *and* $\mathbf{v} = \mathbf{O}\mathbf{b}$. *Then,*

$$\mathrm{E}\left[u_k^2 v_l^2\right] = \begin{cases} \frac{1 + 2\cos^2 \varphi}{n(n+2)} & k = l \\ \frac{n+1 - 2\cos^2 \varphi}{(n-1)n(n+2)} & k \neq l \end{cases} \tag{48}$$

**Proof.** Without loss of generality, the coordinate system can be rotated in such manner that $\mathbf{a} = \mathbf{e}_1$ and $\mathbf{b} = \mathbf{e}_1 \cos \varphi + \mathbf{e}_2 \sin \varphi$. Then, we have

$$\mathbf{u} = \mathbf{o}_1, \tag{49}$$
$$\mathbf{v} = \mathbf{o}_1 \cos \varphi + \mathbf{o}_2 \sin \varphi. \tag{50}$$

For $k = l$, we have

$$\mathrm{E}\left[u_k^2 v_k^2\right] = \mathrm{E}\left[o_{k1}^2 (o_{k1} \cos \varphi + o_{k2} \sin \varphi)^2\right] \tag{51}$$
$$= \mathrm{E}\left[o_{k1}^4\right] \cos^2 \varphi + \mathrm{E}\left[o_{k1}^2 o_{k2}^2\right] \sin^2 \varphi + 2\mathrm{E}\left[o_{k1}^3 o_{k2}\right] \cos \varphi \sin \varphi \tag{52}$$

The last term vanishes because the integral of odd powers of $o_{ij}$ over $\mathcal{S}_n$ is zero. By invoking Lemma 4, we arrive at

$$\mathrm{E}\left[u_k^2 v_k^2\right] = \frac{3\cos^2\varphi}{n(n+2)} + \frac{\sin^2\varphi}{n(n+2)} = \frac{1+2\cos^2\varphi}{n(n+2)} \tag{53}$$

For $k \neq l$,

$$\mathrm{E}\left[u_k^2 v_l^2\right] = \mathrm{E}\left[o_{k1}^2(o_{l1}\cos\varphi + o_{l2}\sin\varphi)^2\right] \tag{54}$$
$$= \mathrm{E}\left[o_{k1}^2 o_{l1}^2\right]\cos^2\varphi + \mathrm{E}\left[o_{k1}^2 o_{l2}^2\right]\sin^2\varphi$$
$$+ 2\mathrm{E}\left[o_{k1}^2 o_{l1} o_{l2}\right]\cos\varphi\sin\varphi \tag{55}$$

The last term vanishes due to odd powers of $o_{ij}$. Together with Lemma 4, we have

$$\mathrm{E}\left[u_k^2 v_l^2\right] = \frac{\cos^2\varphi}{n(n+2)} + \frac{(n+1)\sin^2\varphi}{(n-1)n(n+2)} = \frac{n+1-2\cos^2\varphi}{(n-1)n(n+2)}. \tag{56}$$

□

**Lemma 6.** *Let $\{\mathbf{d}_1,\ldots,\mathbf{d}_{N+1}\}$ be a prototype set of vectors satisfying $\sum_{i=1}^{N+1}\alpha_i\mathbf{d}_i = 0$, where all $\alpha_i \geq 0$ and at least one $\alpha_i \neq 0$. Let $\mathbf{O}$ be a uniformly distributed random orthogonal matrix, and let $\mathbf{v}_i = \mathbf{O}\mathbf{d}_i$ with $\|\mathbf{d}_i\|\|\mathbf{d}_j\|\cos\varphi_{ij} = \mathbf{d}_i^\mathsf{T}\mathbf{d}_j = \mathbf{v}_i^\mathsf{T}\mathbf{v}_j$. Then, the MFN update formula from Lemma 2 involving $N+2$ points ($\mathbf{x}_0$ and the additional $N+1$ points constructed according to (47)) satisfies*

$$\mathrm{E}\left[\|\mathbf{B}_+ - \mathbf{B}\|_\mathrm{F}^2\right] = \mathrm{E}\left[\beta^2\|\mathbf{A}\|_\mathrm{F}^2\right] = (\gamma_1 - \gamma_2)\|\mathbf{B} - \mathbf{H}\|_\mathrm{F}^2 + \gamma_2\mathrm{tr}(\mathbf{B}-\mathbf{H})^2 \tag{57}$$

*where*

$$\gamma_1 = \frac{\mu+2}{n(n+2)} \tag{58}$$

$$\gamma_2 = \frac{(n+1)\mu-2}{(n-1)n(n+2)} \tag{59}$$

$$\mu = \left(\frac{\sum_{i=1}^{N+1}\alpha_i\|\mathbf{d}_i\|^2}{2\|\mathbf{A}\|_\mathrm{F}}\right)^2 = \frac{\left(\sum_{i=1}^{N+1}\alpha_i\|\mathbf{d}_i\|^2\right)^2}{\sum_{i=1}^{N+1}\sum_{j=1}^{N+1}\alpha_i\alpha_j\|\mathbf{d}_i\|^2\|\mathbf{d}_j\|^2\cos^2\varphi_{ij}} \tag{60}$$

**Proof.** By repeating the reasoning in the proof of Lemma 2 on (18), we obtain

$$\mathbf{A}:\mathbf{H} = \sum_{i=1}^{N+1}\alpha_i(q_i - q_0) \tag{61}$$

which yields together with the expression for $\beta$ from Lemma 2

$$\beta = \frac{(\mathbf{H}-\mathbf{B}):\mathbf{A}}{\mathbf{A}:\mathbf{A}} = \frac{\sum_{i=1}^{N+1}\alpha_i\mathbf{v}_i^\mathsf{T}(\mathbf{H}-\mathbf{B})\mathbf{v}_i}{2\|\mathbf{A}\|_\mathrm{F}^2}. \tag{62}$$

Now we can express

$$\beta^2\|\mathbf{A}\|_\mathrm{F}^2 = \frac{\left(\sum_{i=1}^{N+1}\alpha_i\mathbf{v}_i^\mathsf{T}(\mathbf{B}-\mathbf{H})\mathbf{v}_i\right)^2}{4\|\mathbf{A}\|_\mathrm{F}^2}. \tag{63}$$

Because vectors $\mathbf{v}_i$ are uniformly distributed on the unit sphere, we can rotate the coordinate system without affecting $\mathrm{E}\big[\beta^2\|\mathbf{A}\|_{\mathrm{F}}^2\big]$ so that $\mathbf{B} - \mathbf{H}$ is diagonalized.

$$\mathrm{E}\Big[\beta^2\|\mathbf{A}\|_{\mathrm{F}}^2\Big] = \mathrm{E}\left[\frac{\Big(\sum_{i=1}^{N+1}\alpha_i\mathbf{v}_i^{\mathrm{T}}\mathbf{D}\mathbf{v}_i\Big)^2}{4\|\mathbf{A}\|_{\mathrm{F}}^2}\right]. \tag{64}$$

Let $v_{ik}$ denote the $k$-th component of vector $\mathbf{v}_i$ and $\lambda_k$ the $k$-th eigenvalue of $\mathbf{B} - \mathbf{H}$ ($k$-th diagonal element of $\mathbf{D}$).

$$\mathrm{E}\Big[\beta^2\|\mathbf{A}\|_{\mathrm{F}}^2\Big] = \mathrm{E}\left[\frac{\sum_{i=1}^{N+1}\sum_{j=1}^{N+1}\alpha_i\alpha_j\sum_{k=1}^{n}\sum_{l=1}^{n}v_{ik}^2 v_{jl}^2\lambda_k\lambda_l}{4\|\mathbf{A}\|_{\mathrm{F}}^2}\right]. \tag{65}$$

We can rewrite (65) as

$$\mathrm{E}\Big[\beta^2\|\mathbf{A}\|_{\mathrm{F}}^2\Big] = \frac{\sum_{k=1}^{n}\sum_{i=1}^{N+1}\sum_{j=1}^{N+1}\alpha_i\alpha_j\mathrm{E}_{ikjk}\lambda_k^2 + \sum_{k\neq l}\sum_{i=1}^{N+1}\sum_{j=1}^{N+1}\alpha_i\alpha_j\mathrm{E}_{ikjl}\lambda_k\lambda_l}{4\|\mathbf{A}\|_{\mathrm{F}}^2} \tag{66}$$

The expected value of $v_{ik}^2 v_{jl}^2$ depends on $\cos\varphi_{ij} = \mathbf{v}_i^{\mathrm{T}}\mathbf{v}_j$. From Lemma 5, we have

$$\mathrm{E}_{ikjl} = \mathrm{E}\Big[v_{ik}^2 v_{jl}^2\Big] = \begin{cases} \dfrac{1+2\cos^2\varphi_{ij}}{n(n+2)}\|\mathbf{v}_i\|^2\|\mathbf{v}_j\|^2 & k = l \\[2mm] \dfrac{n+1-2\cos^2\varphi_{ij}}{(n-1)n(n+2)}\|\mathbf{v}_i\|^2\|\mathbf{v}_j\|^2 & k \neq l \end{cases} \tag{67}$$

Because the eigenvalues of $\mathbf{D}$ are the same as the eigenvalues of $\mathbf{B} - \mathbf{H}$, we have

$$\|\mathbf{B} - \mathbf{H}\|_{\mathrm{F}}^2 \quad = \quad \|\mathbf{D}\|_{\mathrm{F}}^2 = \sum_{k=1}^{n}\lambda_k^2, \tag{68}$$

$$\mathrm{tr}(\mathbf{B} - \mathbf{H})^2 \quad = \quad \mathrm{tr}(\mathbf{D})^2 = \left(\sum_{k=1}^{n}\lambda_k\right)^2 = \sum_{k=1}^{n}\sum_{l=1}^{n}\lambda_k\lambda_l \tag{69}$$

and

$$(\gamma_1 - \gamma_2)\|\mathbf{B} - \mathbf{H}\|_{\mathrm{F}}^2 + \gamma_2\mathrm{tr}(\mathbf{B} - \mathbf{H})^2 \quad = \quad (\gamma_1 - \gamma_2)\sum_{k=1}^{n}\lambda_k^2 + \gamma_2\sum_{k=1}^{n}\sum_{l=1}^{n}\lambda_k\lambda_l$$

$$= \quad \gamma_1\sum_{k=1}^{n}\lambda_k^2 + \gamma_2\sum_{k\neq l}\lambda_k\lambda_l \tag{70}$$

Note that $\|\mathbf{v}_i\| = \|\mathbf{d}_i\|$. Taking into account (66), (67), and (70) yields

$$\gamma_1 \quad = \quad \frac{\sum_{i=1}^{N+1}\sum_{j=1}^{N+1}\alpha_i\alpha_j\|\mathbf{d}_i\|^2\|\mathbf{d}_j\|^2\big(1 + 2\cos^2\varphi_{ij}\big)}{4n(n+2)\|\mathbf{A}\|_{\mathrm{F}}^2}, \tag{71}$$

$$\gamma_2 \quad = \quad \frac{\sum_{i=1}^{N+1}\sum_{j=1}^{N+1}\alpha_i\alpha_j\|\mathbf{d}_i\|^2\|\mathbf{d}_j\|^2\big(n + 1 - 2\cos^2\varphi_{ij}\big)}{4(n-1)n(n+2)\|\mathbf{A}\|_{\mathrm{F}}^2}. \tag{72}$$

The Frobenius norm of $\mathbf{A}$ can be expressed as

$$\begin{aligned}\|\mathbf{A}\|_{\mathrm{F}}^2 &= \mathrm{tr}(\mathbf{A}^{\mathrm{T}}\mathbf{A}) = \mathrm{tr}\left(\Big(\tfrac{1}{2}\sum_{i=1}^{N+1}\alpha_i\mathbf{v}_i\mathbf{v}_i^{\mathrm{T}}\Big)^{\mathrm{T}}\Big(\tfrac{1}{2}\sum_{j=1}^{N+1}\alpha_j\mathbf{v}_j\mathbf{v}_j^{\mathrm{T}}\Big)\right) \\ &= \tfrac{1}{4}\sum_{i=1}^{N+1}\sum_{j=1}^{N+1}\alpha_i\alpha_j\mathrm{tr}\Big(\mathbf{v}_i\mathbf{v}_i^{\mathrm{T}}\mathbf{v}_j\mathbf{v}_j^{\mathrm{T}}\Big) = \tfrac{1}{4}\sum_{i=1}^{N+1}\sum_{j=1}^{N+1}\alpha_i\alpha_j\big(\mathbf{v}_i^{\mathrm{T}}\mathbf{v}_j\big)^2 \\ &= \tfrac{1}{4}\sum_{i=1}^{N+1}\sum_{j=1}^{N+1}\alpha_i\alpha_j\|\mathbf{d}_i\|^2\|\mathbf{d}_j\|^2\cos^2\varphi_{ij}\end{aligned} \tag{73}$$

By substituting (73) in (71) and (72), we arrive at

$$\gamma_1 = \frac{8\|\mathbf{A}\|_{\mathrm{F}}^2 + \sum_{i=1}^{N+1}\sum_{j=1}^{N+1} \alpha_i\alpha_j\|\mathbf{d}_i\|^2\|\mathbf{d}_j\|^2}{4n(n+2)\|\mathbf{A}\|_{\mathrm{F}}^2}, \tag{74}$$

$$\gamma_2 = \frac{(n+1)\sum_{i=1}^{N+1}\sum_{j=1}^{N+1} \alpha_i\alpha_j\|\mathbf{d}_i\|^2\|\mathbf{d}_j\|^2 - 8\|\mathbf{A}\|_{\mathrm{F}}^2}{4(n-1)n(n+2)\|\mathbf{A}\|_{\mathrm{F}}^2}. \tag{75}$$

We also have

$$\sum_{i=1}^{N+1}\sum_{j=1}^{N+1} \alpha_i\alpha_j\|\mathbf{d}_i\|^2\|\mathbf{d}_j\|^2 = \left(\sum_{i=1}^{N+1} \alpha_i\|\mathbf{d}_i\|^2\right)^2 \tag{76}$$

Substituting (76) into (74) and (75) concludes the proof. □

**Theorem 1.** *Let $\gamma_1$, $\gamma_2$, and $\mu$ be defined as in Lemma 6. Then,*

$$\frac{\mathrm{E}\big[\|\mathbf{B}_+ - \mathbf{H}\|_{\mathrm{F}}^2\big]}{\|\mathbf{B} - \mathbf{H}\|_{\mathrm{F}}^2} \leq \begin{cases} 1 - \frac{2(n-\mu)}{(n-1)n(n+2)} & \mu \geq \frac{2}{n+1} \\ 1 - \frac{\mu}{n} & \mu < \frac{2}{n+1} \end{cases}. \tag{77}$$

**Proof.** We start with the following identity.

$$\mathbf{B}_+ - \mathbf{H} + \mathbf{B} - \mathbf{B}_+ = \mathbf{B} - \mathbf{H}. \tag{78}$$

Computing the Frobenius norm on both sides and considering $(\mathbf{B}_+ - \mathbf{B}) \perp (\mathbf{B}_+ - \mathbf{H})$ results in

$$\|\mathbf{B}_+ - \mathbf{H}\|_{\mathrm{F}}^2 + \|\mathbf{B}_+ - \mathbf{B}\|_{\mathrm{F}}^2 = \|\mathbf{B} - \mathbf{H}\|_{\mathrm{F}}^2. \tag{79}$$

Taking into account (15) results in

$$\|\mathbf{B}_+ - \mathbf{H}\|_{\mathrm{F}}^2 = \|\mathbf{B} - \mathbf{H}\|_{\mathrm{F}}^2 - \beta^2\|\mathbf{A}\|_{\mathrm{F}}^2. \tag{80}$$

After Lemma 6 is applied, we have

$$\frac{\mathrm{E}\big[\|\mathbf{B}_+ - \mathbf{H}\|_{\mathrm{F}}^2\big]}{\|\mathbf{B} - \mathbf{H}\|_{\mathrm{F}}^2} = 1 - \left(\gamma_1 - \gamma_2 + \gamma_2 \frac{\mathrm{tr}(\mathbf{B} - \mathbf{H})^2}{\|\mathbf{B} - \mathbf{H}\|_{\mathrm{F}}^2}\right) \tag{81}$$

By definition, $\mu \geq 0$ and $\gamma_1 \geq 0$. For $\gamma_2 \geq 0$, we must have $\mu \geq 2/(n+1)$. By considering $\mathrm{tr}(\mathbf{B} - \mathbf{H})^2/\|\mathbf{B} - \mathbf{H}\|_{\mathrm{F}}^2 \geq 0$, we arrive at

$$\frac{\mathrm{E}\big[\|\mathbf{B}_+ - \mathbf{H}\|_{\mathrm{F}}^2\big]}{\|\mathbf{B} - \mathbf{H}\|_{\mathrm{F}}^2} \leq 1 - (\gamma_1 - \gamma_2) = 1 - \frac{2(n-\mu)}{(n-1)n(n+2)}. \tag{82}$$

For $\gamma_2 < 0$, we must have $\mu < 2/(n+1)$. Invoking Lemma A1 yields

$$\frac{\mathrm{E}\big[\|\mathbf{B}_+ - \mathbf{H}\|_{\mathrm{F}}^2\big]}{\|\mathbf{B} - \mathbf{H}\|_{\mathrm{F}}^2} \leq 1 - (\gamma_1 - \gamma_2 + n\gamma_2) = 1 - (\gamma_1 + (n-1)\gamma_2) = 1 - \frac{\mu}{n}. \tag{83}$$

□

From Theorem 1, several results can be derived. First, we will assume the prototype set is a regular $N$-simplex (i.e., comprises $N + 1$ vectors positively spanning an $N$-dimensional subspace). This case is interesting because the update formula in [9] is obtained for $N = 1$. We are going to show that our estimate of the expected Hessian improvement is identical to the one published in [9]. This update formula (with $N = 1$) was used in an optimization algorithm published in [10].

Next, we are going to show that using a regular $n$-simplex as the prototype set is a bad choice. According to Theorem 1, no improvement of the Hessian is guaranteed. Even worse, we show that improvement occurs only at the first application of the update formula.

Finally, we will analyze the case where the prototype set is what we refer to as an augmented set of $N$ orthonormal vectors. Such a prototype set with $N = n$ was used in the optimization algorithm published in [14].

**Corollary 1.** *Let $\mathcal{D}$ be a regular $N$-simplex ($N \leq n$). Then,*

$$\frac{\mathrm{E}\left[\|\mathbf{B}_+ - \mathbf{H}\|_{\mathrm{F}}^2\right]}{\|\mathbf{B} - \mathbf{H}\|_{\mathrm{F}}^2} \leq 1 - \frac{2(n - N)}{(n - 1)n(n + 2)} \tag{84}$$

**Proof.** For all $i \neq j$, we have $\cos \varphi_{ij} = -N^{-1}$ and $\|\mathbf{d}_i\| = 1$. Because the sum of all vectors in a regular $N$-simplex is $\mathbf{0}$, we conclude $\alpha_i = 1$ and

$$\mu = \frac{\left(\sum_{i=1}^{N+1} \alpha_j\right)^2}{\sum_{i=1}^{N+1} \sum_{j=1}^{N+1} \alpha_i \alpha_j \cos^2 \varphi_{ij}} = \frac{(N + 1)^2}{(N + 1) \cdot 1 + N(N + 1) \cdot N^{-2}} = N \tag{85}$$

Because $2/(n + 1) \leq 1$ for all $n \geq 1$, we have $\mu \geq 2/(n + 1)$, and the result follows from Theorem 1. $\square$

Corollary 1 implies that the most efficient approach to MFN updating with a regular simplex in the role of the prototype set of unit vectors is to use a regular 1-simplex (three collinear points).

**Corollary 2.** *For $N = 1$ and $\mathbf{d}_1 = -\mathbf{d}_2$, set $\mathcal{D}$ is a regular 1-simplex and*

$$\frac{\mathrm{E}\left[\|\mathbf{B}_+ - \mathbf{H}\|_{\mathrm{F}}^2\right]}{\|\mathbf{B} - \mathbf{H}\|_{\mathrm{F}}^2} \leq 1 - \frac{2}{n(n + 2)}. \tag{86}$$

This result was proven in [9] with a less general approach. Here, we obtain it as a special case of Corollary 1 for $N = 1$.

According to Corollary 1, there is no guaranteed improvement of $\|\mathbf{B} - \mathbf{H}\|_{\mathrm{F}}$ if a regular $n$-simplex ($N = n$) is used in the update process. In fact, the situation is even worse as we show in the following Lemma.

**Lemma 7.** *If $\mathcal{D}$ is a regular $n$-simplex ($N = n$), then the MFN update from Lemma 2 improves the Hessian approximation only in its first application.*

**Proof.** From Lemma 2, we can see that

$$\mathbf{B}_+ = \mathbf{B} + \beta \mathbf{A} \tag{87}$$

where (see (62))

$$\beta = \frac{\sum_{i=1}^{N+1} \alpha_i \mathbf{v}_i^{\mathrm{T}} (\mathbf{H} - \mathbf{B}) \mathbf{v}_i}{2\|\mathbf{A}\|_{\mathrm{F}}^2}. \tag{88}$$

For a regular simplex, $\alpha_i = 1$ and $\|\mathbf{d}_i\| = 1$. The Frobenius norm of $\mathbf{A}$ is

$$\|\mathbf{A}\|_{\mathrm{F}}^2 = \frac{1}{4} \sum_{i=1}^{N+1} \sum_{j=1}^{N+1} \cos^2 \varphi_{ij} = \frac{1}{4} \cdot \left(1 \cdot (n + 1) + \frac{1}{n^2} \cdot n(n + 1)\right) = \frac{(n + 1)^2}{4n} \tag{89}$$

Due to Lemma A3 (see Appendix A for proof), we have

$$\sum_{i=1}^{N+1} \alpha_i \mathbf{v}_i^{\mathrm{T}}(\mathbf{H} - \mathbf{B})\mathbf{v}_i = \frac{n+1}{n}\mathrm{tr}(\mathbf{H} - \mathbf{B}) \tag{90}$$

and

$$\beta = \frac{2}{n+1}\mathrm{tr}(\mathbf{H} - \mathbf{B}) \tag{91}$$

From definition of $\mathbf{A}$, we obtain

$$\mathrm{tr}(\mathbf{A}) = \frac{1}{2}\sum_{i=1}^{n+1} \alpha_i \mathrm{tr}\left(\mathbf{v}_i \mathbf{v}_i^{\mathrm{T}}\right) = \frac{1}{2}\sum_{i=1}^{n+1} \alpha_i \|\mathbf{v}_i\|^2 = \frac{n+1}{2} \tag{92}$$

From $\mathbf{B}_+ - \mathbf{H} = \mathbf{B} - \mathbf{H} + \beta\mathbf{A}$, we can express

$$\mathrm{tr}(\mathbf{B}_+ - \mathbf{H}) = \mathrm{tr}(\mathbf{B} - \mathbf{H}) + \beta\mathrm{tr}(\mathbf{A}) = 0. \tag{93}$$

Let $\mathbf{B}_{++}$ denote the approximate Hessian after the second application of the update formula.

$$\mathbf{B}_{++} = \mathbf{B}_+ + \beta_+ \mathbf{A}_+ \tag{94}$$

Because $\beta_+ = 2\mathrm{tr}(\mathbf{H} - \mathbf{B}_+)/(n+1) = 0$, we have $\mathbf{B}_{++} = \mathbf{B}_+$, and the proof is complete. □

Intuition can mislead one into considering the regular $n$-simplex as the best choice for positioning $n+1$ points around an origin $\mathbf{x}_0$ when computing an MFN update based on Lemma 2. Lemma 7 shows the exact opposite—a regular $n$-simplex is the worst choice because the update formula does not improve the Hessian approximation in its second and all subsequent applications.

**Definition 2.** *An augmented set of $1 \leq N \leq n$ orthonormal vectors is a set comprising $N$ mutually orthogonal unit vectors $\mathbf{e}_1, \ldots, \mathbf{e}_N$ and their normalized negative sum $-N^{-1/2}(\mathbf{e}_1 + \ldots + \mathbf{e}_N)$.*

Note that an augmented set of $N = 1$ orthonormal vectors is equivalent to a regular 1-simplex. Now, we have $\|\mathbf{d}_i\| = 1$ and $\cos\varphi_{ii} = 1$. For $i \neq j$, we have $\cos\varphi_{ij} = 0$ except for $i = n+1$ or $j = n+1$ when $\cos\varphi_{ij} = -N^{-1/2}$. Because $\mathbf{d}_{N+1}$ is the normalized negative sum of the first $N$ vectors, $\alpha_1 = \ldots = \alpha_n = 1$ and $\alpha_{n+1} = N^{1/2}$.

**Corollary 3.** *If the prototype set of unit vectors is an augmented set of $N$ orthonormal vectors, then*

$$\frac{\mathrm{E}\left[\|\mathbf{B}_+ - \mathbf{H}\|_{\mathrm{F}}^2\right]}{\|\mathbf{B} - \mathbf{H}\|_{\mathrm{F}}^2} \leq 1 - \frac{2n - N - N^{1/2}}{(n-1)n(n+2)}. \tag{95}$$

**Proof.**

$$\mu = \frac{\left(\sum_{j=1}^{N+1} \alpha_i\right)^2}{\sum_{i=1}^{N+1}\sum_{j=1}^{N+1} \alpha_i \alpha_j \cos^2\varphi_{ij}} = \frac{\left(N \cdot 1 + N^{1/2}\right)^2}{N \cdot 1 \cdot 1 \cdot 1^2 + 1 \cdot N^{1/2} \cdot N^{1/2} \cdot 1^2 + 2N \cdot 1 \cdot N^{1/2} \cdot N^{-1}}$$

$$= \frac{N(N^{1/2}+1)^2}{2N^{1/2}(N^{1/2}+1)} = \frac{N + N^{1/2}}{2} \tag{96}$$

Because $2/(n+1) \leq 1$ for all $n \geq 1$, we conclude $\mu \geq 2/(n+1)$, and the result follows from Theorem 1. □

A special case of Corollary 1 is the following result.

**Corollary 4.** *If the prototype set of unit vectors is an augmented set of $N = n$ orthonormal vectors, then*

$$\frac{\mathrm{E}\left[\|\mathbf{B}_+ - \mathbf{H}\|_{\mathrm{F}}^2\right]}{\|\mathbf{B} - \mathbf{H}\|_{\mathrm{F}}^2} \leq 1 - \frac{1}{(n + n^{1/2})(n + 2)}. \tag{97}$$

Corollaries 2 and 4 indicate that for an augmented set of $N = n$ orthonormal vectors (used in [12]), the expected improvement of the approximate Hessian approaches half of the improvement obtained using a regular 1-simplex (introduced in [9]) when $n$ approaches infinity.

Corollaries 1–4 indicate that the update formula yields a greater improvement of the approximate Hessian when the prototype set of vectors exhibits more directionality, in the sense that the vectors are confined to an $N < n$ dimensional subspace of the search space. Lower values of $N$ result in faster convergence.

## 5. Example

We illustrate the proposed update with a simple example. The sequence of uniformly distributed orthogonal matrices is generated as in [14]. Three prototype sets are examined—regular 1-dimensional and $(n-1)$-dimensional simplex and the augmented set of $n$ orthonormal vectors. The true Hessian $\mathbf{H}$ is chosen randomly and the initial Hessian approximation is set to $\mathbf{B} = \mathbf{0}$. The progress of the update is measured by the normalized Frobenius distance between $\mathbf{H}$ and $\mathbf{B}$.

Figure 1 depicts the progress of the proposed update with various prototype sets for $n = 5$ and $n = 10$. It is clearly visible that the convergence of the update is linear and depends on the choice of the prototype set. The convergence rate of the update using an augmented set of $n$ orthonormal vectors is approximately half of the convergence rate exhibited by the update using a regular 1-simplex. It can also be seen that the bound on the amount of progress obtained from one update (Theorem 1) is fairly conservative. The actual progress of the update is much better in practice.

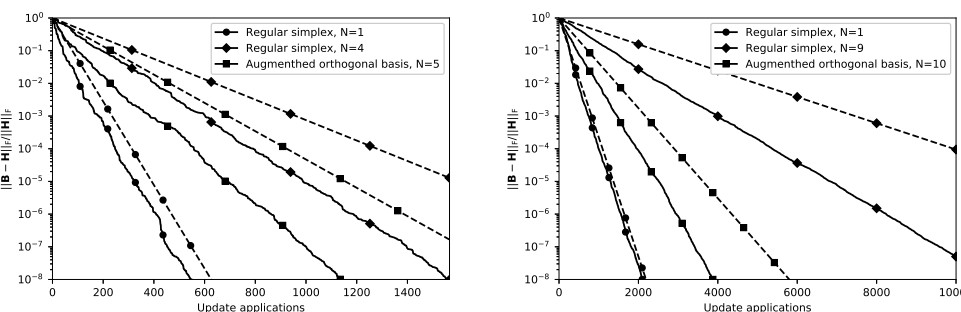

**Figure 1.** Progress of three simplicial updates for $n = 5$ (**left**) and $n = 10$ (**right**). Dashed lines represent the progress of the update assuming every update application improves the approximate Hessian by the amount predicted in Theorem 1.

## 6. Discussion

The convergence of a Hessian update formula that requires only function values for computing the update was analyzed. The update formula is based on the formula published in [8] that generally requires the function values at $m \geq n + 2$ points. The proposed update is based on the case where $m = n + 2$. An additional requirement is introduced, namely that the $m - 1$ vectors from the central point to the remaining $m - 1$ points must positively span a $m - 2$ dimensional subspace of $\mathbb{R}^n$. This requirement extends the usability of the proposed update to sets of points with $3 \leq m \leq n + 2$ members. The set of $m$ points used by the update is generated by adding $m - 1$ vectors to a central point in the set. The vectors are obtained by applying a random orthogonal transformation to a prototype set of vectors that spans a $m - 2$ dimensional subspace of $\mathbb{R}^n$.

A lower bound on the expected improvement of the Hessian approximation was derived (Theorem 1). Up to now, no such result was published for the update from [8] and $m = n + 2$. The obtained result was applied to several different prototype sets. The general result obtained for the case when the prototype set is a regular $m − 2$-dimensional simplex (Corollary 1) shows that the expected improvement of the Hessian approximation is greatest for $m = 3$ (i.e., 1-dimensional regular simplex) and decreases as the dimensionality of the simplex increases. The special case when $m = 3$ (1-dimensional regular simplex) corresponds to the update from [9]. The lower bound on the expected improvement obtained with our general result (Corollary 2) matches the one that was published in [9]. For the $n$-dimensional regular simplex, our result indicates that the lower bound on expected improvement of the Hessian approximation is 0. Furthermore, it was shown that the Hessian approximation is possibly improved only by the first application of the proposed update formula (Lemma 7). Therefore, the use of the $n$-dimensional regular simplex in the role of the prototype set is a bad choice.

Next, the expected improvement of the approximate Hessian for a prototype set comprising $N \leq n$ orthogonal vectors and their normalized negative sum was derived. Such a prototype set with $N = n$ was used in the optimization algorithm published in [12]. It was shown that using this kind of prototype set does guarantee a positive lower bound on the expected improvement of the Hessian approximation (Corollary 4). The general result (Corollary 3), however, again indicates that using a prototype set of lower dimensionality results in faster convergence. The result for $N = 1$ (two collinear vectors in the role of the prototype set) is the same as the one obtained for the update from [9].

Finally, the results were illustrated by running the proposed update on a quadratic function with a randomly chosen Hessian for several choices of the prototype set. The observed progress was compared to the lower bound predicted by Theorem 1. The results indicate that the lower bound is quite pessimistic, and that the actual progress is faster. The observed performance was closest to the predicted lower bound for the update formula from [9].

**Author Contributions:** Conceptualization, Á.B. and T.T.; methodology, Á.B. and J.O.; software, J.O.; validation, J.O.; formal analysis, Á.B.; investigation, Á.B.; resources, T.T.; data curation, Á.B.; writing—original draft preparation, Á.B.; writing—review and editing, Á.B. and J.O.; visualization, J.O.; supervision, T.T.; project administration, T.T.; funding acquisition, T.T. All authors have read and agreed to the published version of the manuscript.

**Funding:** The research was co-funded by the Ministry of Education, Science, and Sport (Ministrstvo za Šolstvo, Znanost in Šport) of the Republic of Slovenia through the program P2-0246 ICT4QoL—Information and Communications Technologies for Quality of Life.

**Institutional Review Board Statement:** Not applicable.

**Informed Consent Statement:** Not applicable.

**Data Availability Statement:** Not applicable.

**Acknowledgments:** The authors would like to thank the anonymous referees for their useful comments that helped to improve the paper. Most notably, the authors would like to thank the second referee whose suggestion lead to the simplification of the proof of Lemma 4.

**Conflicts of Interest:** The authors declare no conflict of interest.

## Abbreviations

The following abbreviations are used in this manuscript:

| | |
|---|---|
| MFN | Minimum Frobenius norm |
| BFGS | Broyden-Fletcher-Goldfarb-Shanno |
| SR1 | Symmetric rank-one |

**Appendix A**

The following lemma is used in the proof of the main result.

**Lemma A1.** *Let* **B** *be a* $n \times n$ *matrix. Then,*

$$\text{tr}(\mathbf{B})^2 \leq n\|\mathbf{B}\|_{\text{F}}^2 \tag{A1}$$

**Proof.** Let $\lambda_i \in \mathbb{R}$ denote the $n$ eigenvalues of **B**. We have

$$\text{tr}(\mathbf{B})^2 = \left(\sum_{i=1}^{n} \lambda_i\right)^2, \tag{A2}$$

$$\|\mathbf{B}\|_{\text{F}}^2 = \sum_{i=1}^{n} \lambda_i^2 = a. \tag{A3}$$

The maximum of $\text{tr}(\mathbf{B})^2$ can be obtained by finding $\max\left(\sum_{i=1}^{n} \lambda_i\right)^2$ subject to $\sum_{i=1}^{n} \lambda_i^2 = a$. The solution of this problem is

$$|\lambda_i| = (a/n)^{1/2} \quad i = 1, 2, \ldots, n. \tag{A4}$$

Considering $\sum_{i=1}^{n} \lambda_i \leq n(a/n)^{1/2}$ along with (A2) and (A3) concludes the proof. $\square$

Let **S** be a $n \times (n+1)$ matrix whose columns are the vectors comprising a regular simplex in $n$ dimensions. By definition, the following must hold

$$\mathbf{S}^{\text{T}}\mathbf{S} = \begin{bmatrix} 1 & -n^{-1} & \ldots & -n^{-1} \\ -n^{-1} & 1 & \ldots & -n^{-1} \\ \vdots & \vdots & \ddots & \vdots \\ -n^{-1} & -n^{-1} & \ldots & 1 \end{bmatrix}_{(n+1)\times(n+1)} = \mathbf{C} \tag{A5}$$

Clearly, there are infinitely many possible solutions to (A5). We will assume that **S** is upper triangular. A solution to (A5) with this property is unique and can be obtained via Cholesky decomposition of the submatrix of **C** comprising the first $n$ rows and columns which yields the first $n$ columns of **S**. The last column is then obtained as the negative sum of the first $n$ columns. Matrix **S** is in row echelon form and represents what we will refer to as the standard regular simplex. Its components can be expressed as

$$s_{ii}^2 = \frac{(n+1)(n-i+1)}{n(n-i+2)} \tag{A6}$$

$$s_{ij} = \begin{cases} -\frac{s_{ii}}{n-i+1} & j > i \\ 0 & \text{otherwise} \end{cases} \tag{A7}$$

**Lemma A2.** *Let columns of* **V** *represent a regular simplex. Then,*

$$\mathbf{V}\mathbf{V}^{\text{T}} = \mathbf{S}\mathbf{S}^{\text{T}} = \frac{n+1}{n}\mathbf{I}_{n\times n} \tag{A8}$$

**Proof.** Let columns of **S** comprise a standard regular simplex. Diagonal elements of $\mathbf{S}\mathbf{S}^{\text{T}}$ can be obtained as

$$\sum_{i=1}^{n+1} s_{ki}^2 = \sum_{i=k}^{n+1} s_{ki}^2 = s_{kk}^2 + (n-k+1)s_{k(k+1)}^2 = s_{kk}^2 \cdot \frac{n-k+2}{n-k+1} = \frac{n+1}{n}. \tag{A9}$$

Because $\mathbf{SS}^{\mathrm{T}}$ is symmetric, we assume $k > l$ for computing the extradiagonal elements

$$\sum_{i=1}^{n+1} s_{ki}s_{li} = \sum_{i=k}^{n+1} s_{ki}s_{li} = s_{l(l+1)} \sum_{i=k}^{n+1} s_{ki} = s_{l(l+1)}\left(s_{kk} + (n-k+1)\cdot s_{k(k+1)}\right) \quad \text{(A10)}$$

$$= s_{l(l+1)}\left(s_{kk} - (n-k+1)\frac{s_{kk}}{n-k+1}\right) = 0 \quad \text{(A11)}$$

This proves $\mathbf{SS}^{\mathrm{T}} = (n+1)\mathbf{I}_{n\times n}/n$. Any regular simplex $\mathbf{V}$ can be expressed with the standard regular simplex as $\mathbf{V} = \mathbf{QS}$, where $\mathbf{Q}$ is an orthogonal matrix. Therefore, we have

$$\mathbf{VV}^{\mathrm{T}} = \mathbf{QSS}^{\mathrm{T}}\mathbf{Q}^{\mathrm{T}} = \frac{n+1}{n}\mathbf{QI}_{n\times n}\mathbf{Q}^{\mathrm{T}} = \frac{n+1}{n}\mathbf{I}_{n\times n} \quad \text{(A12)}$$

□

**Lemma A3.** *Let columns of* $\mathbf{V}$ *represent a regular simplex, and let* $\mathbf{H}$ *be a symmetric matrix. Then,*

$$\sum_{i=1}^{n+1} \mathbf{v}_i^{\mathrm{T}}\mathbf{H}\mathbf{v}_i = \frac{n+1}{n}\mathrm{tr}(\mathbf{H}) \quad \text{(A13)}$$

**Proof.**

$$\sum_{i=1}^{n+1} \mathbf{v}_i^{\mathrm{T}}\mathbf{H}\mathbf{v}_i = \mathrm{tr}(\mathbf{V}^{\mathrm{T}}\mathbf{H}\mathbf{V}) = \mathrm{tr}\left(\mathbf{H} : \left(\mathbf{VV}^{\mathrm{T}}\right)\right) = \frac{n+1}{n}\mathrm{tr}(\mathbf{H} : \mathbf{I}) = \frac{n+1}{n}\sum_{i=1}^{n} h_{ii} \quad \text{(A14)}$$

□

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
