# Peer review of "Randomized Simplicial Hessian Update"

_mathematics, doi:10.3390/math9151775_

Round 1
Reviewer 1 Report
The paper is technically interesting; however, some minorimprovements are needed before I can recommend publication.
Abstract section should be improved considering the following
structure: introduction, problem statement, methodology, results, and
conclusion.
In Introduction section, the authors should improve the research
background, the review of significant works in the specific study area,
the knowledge gap, the problem statement, and the novelty of the research.
Introduction is not organized well. The reviewer cannot readily see
the significance of this study compared to the previous works. The
literature review should be extended to more recently published works
available in the literature.
Effective optimization techniques can be used to automate generating
models which are normally tedious to do one by one. The authors could
discuss the optimisation techniques used in the papers below, it would
be good for readers to know how optimisation techniques can be used in
real life problems.
(A) Z.Y. Fang, K. Roy, B.S. Chen, C.-W. Sham, I. Hajirasouliha, J.B.P.
Lim, Deep learning-based procedure for structural design of cold-formed
steel channel sections with edge-stiffened and un-stiffened holes under
axial compression, Thin-Walled Struct (2021), Volume 166, September
2021, 108076."
Author Response
> The paper is technically interesting; however, some minor
> improvements are needed before I can recommend publication.
Thank you for your insightful comments. The responses to your comments are listed below.
> Abstract section should be improved considering the following
> structure: introduction, problem statement, methodology, results, and
> conclusion.
The abstract was reorganized and extended.
> In Introduction section, the authors should improve the research
> background, the review of significant works in the specific study area,
> the knowledge gap, the problem statement, and the novelty of the research.
> Introduction is not organized well. The reviewer cannot readily see
> the significance of this study compared to the previous works. The
> literature review should be extended to more recently published works
> available in the literature.
The introduction was extended and now also states the problem definition.
The research is novel. The problem, however, is a niche problem in black-box optimization. Therefore the list of relevant literature is not long. The problem itself deals with the computation of the approximate Hessian of a quadratic model of a function. The approximate Hessian is obtained via an update process (i.e. gradually, just like in the well-known BFGS method). Contrary to the BFGS method which requires the knowledge of the function's gradient, the proposed update formula requires only the function values to be known. This makes it particularly well suited for derivative-free black-box optimization algorithms.
We extended the list of literature with several entries.
> Effective optimization techniques can be used to automate generating
> models which are normally tedious to do one by one. The authors could
> discuss the optimisation techniques used in the papers below, it would
> be good for readers to know how optimisation techniques can be used in
> real life problems.
> (A) Z.Y. Fang, K. Roy, B.S. Chen, C.-W. Sham, I. Hajirasouliha, J.B.P.
> Lim, Deep learning-based procedure for structural design of cold-formed
> steel channel sections with edge-stiffened and un-stiffened holes under
> axial compression, Thin-Walled Struct (2021), Volume 166, September
> 2021, 108076."
Done.
Reviewer 2 Report
This paper provides an in debt analysis of a niche problem in BBO (Black-Box Optimization). In particular the problem where an estimate of the Hessian is pursued based on function evaluations over a set of interpolation points
- provided new interpolation points that were generated by randomly orienting a prototype set
- using the MFN update strategy introduced by Powell
The analysis is sound and the proofs are correct. I belief this analysis provides new results and is of interest to this niche’s community, especially in light of the journal’s topic.
However I also think an attempt should be made to make the content more accessible to a broader audience. Being familiar with BBO I still found it hard to follow precisely what problem was studied until section 4. I wonder why the problem is not stated in advance and why a more elaborate introduction of the problem is not given in the introduction simultaneously framing it into the context of BBO. In particular coming from RBF interpolation based BBO I have a general question regarding this strategy:
- All properties are illustrated on quadratic problems, how do these results relate to the intended use in BBO? The regular simplex is a bad choice in a quadratic landscape but what about a general nonlinear landscape?
I have the following remarks regarding the text and presentation
- It is unclear to this reviewer why is Lemma 1 included.
- If convincing arguments are present to include Lemma 1, some details should be specified that are not trivial, at least to this reviewer
- line 84, Why can we assume that H = 0 without loss of generality?
- line 89, MFN update is geometrically equivalent to finding the ball… without proper details about the MFN update this is unclear
- Although Lemma 2 contains the central result from which all other results are derived, to my opinion the proof can be omitted, or at least there should be a proper reference to the earlier derivation
- I would propose to combine Lemma 4 to 6 in a single Lemma. This reviewer also beliefs the proof of Lemma 5 can be much shorter given properties of O transposed and the result from Lemma 4.
- Lemma 9 seems to be a quite general result. It seems the proof can be omitted and the Lemma itself could be moved to the appendices.
- Line 157 after the proof of Theorem 1 would benefit some additional explanation and a look-out for the different prototype sets that will be analysed using this result.
- Throughout the text I also found some small typos. Please verify with a spelling corrector.
Author Response
> This paper provides an in debt analysis of a niche problem in BBO (Black-Box Optimization). In particular the problem where an estimate of the Hessian is pursued based on function evaluations over a set of interpolation points
>
> provided new interpolation points that were generated by randomly orienting a prototype set
> using the MFN update strategy introduced by Powell
>
> The analysis is sound and the proofs are correct. I belief this analysis provides new results and is of interest to this niche’s community, especially in light of the journal’s topic.
Thank you for your insightful comments. The responses to your comments are listed below.
> However I also think an attempt should be made to make the content more accessible to a broader audience. Being familiar with BBO I still found it hard to follow precisely what problem was studied until section 4. I wonder why the problem is not stated in advance and why a more elaborate introduction of the problem is not given in the introduction simultaneously framing it into the context of BBO.
We added an introduction that frames our formula in the context of BBO. We also added a short overview of how the formula is obtained.
> In particular coming from RBF interpolation based BBO I have a general question regarding this strategy:
>
> All properties are illustrated on quadratic problems, how do these results relate to the intended use in BBO? The regular simplex is a bad choice in a quadratic landscape but what about a general nonlinear landscape?
The paper is about the convergence rate of a method for constructing quadratic models. We analyze the expected improvement of the Hessian approximation (in terms of Frobenius norm) that is obtained with one application of the update formula. We are not solving any particular BBO oproblem. The method we are analyzing could be used in a black box (BB) optimizer (as a matter of fact, it was; see (Bűrmen and Fajfar, COAP 2019)). The derivation of the update and the analysis of its convergence are based on the assumption that the function we are approximating is quadratic. Nevertheless the update can be used with an arbitrary nonlinear function because as the optimizer converges to some point x the points used in the update also converge to the neighborhood of x. Assuming the modeled function is twice continuously differentiable the Hessian approximation approaches the true Hessian as the optimizer converges towards x. This results in significantly faster (usually superlinear) convergence of a BB optimizer in the neighborhood of a minimum. A similar reasoning was used in (Leventhal and Lewis, Opt. 2011).
In our derivation we show that using a regular simplex in the role of a prototype set is not appropriate. Assuming the function we are approximating is not quadratic, the approximate Hessian will still not approach the true Hessian as the BB optimizer converges, because the update is incapable of improving the Hessian approximation when a regular simplex is used in the role of the prototype set.
> I have the following remarks regarding the text and presentation
> It is unclear to this reviewer why is Lemma 1 included.
> If convincing arguments are present to include Lemma 1, some details should be specified that are not trivial, at least to this reviewer
Lemma 1 is used in the derivation of the proposed update formula. A reference to it was added.
> line 84, Why can we assume that H = 0 without loss of generality?
> line 89, MFN update is geometrically equivalent to finding the ball… without proper details about the MFN update this is unclear
We modified the proof of Lemma 1 to resolve these issues. The definition of a MFN update is given in the first paragraph of Section 2.
> Although Lemma 2 contains the central result from which all other results are derived, to my opinion the proof can be omitted, or at least there should be a proper reference to the earlier derivation
Lemma 2 derives the proposed update formula. The formula was first derived in (Bürmen and Fajfar, Comput. Optim. Appl., 2019) where the derivation was based on solving a constrained minimization problem. We believe this approach is more elegant and general, and would therefore like to keep it in the paper.
> I would propose to combine Lemma 4 to 6 in a single Lemma. This reviewer also beliefs the proof of Lemma 5 can be much shorter given properties of O transposed and the result from Lemma 4.
We merged the three Lemmas. Thanks for the hint with transposed O. It results in an elegant proof.
> Lemma 9 seems to be a quite general result. It seems the proof can be omitted and the Lemma itself could be moved to the appendices.
Done. The proof was shortened to a few lines.
> Line 157 after the proof of Theorem 1 would benefit some additional explanation and a look-out for the different prototype sets that will be analysed using this result.
Added 3 paragraphs that announce the results that follow from Theorem 1.
> Throughout the text I also found some small typos. Please verify with a spelling corrector.
We spell checked the document and fixed the errors.
Reviewer 3 Report
The paper is well written and interesting. I recommend the publication of the paper in its present form.
The paper analyzes a technique for the Hessian update in a quadratic function model using function values only. The analysis extends some previous results to the case where the update is constructed from just a few function values. I think the paper is quite interesting and generally well written and, therefore, I recommend publication in its present form. Minor typos: f is not defined at lin e 94.
Author Response
> The paper is well written and interesting. I recommend the publication of the paper in its present form.
>
Thank you for your comments. The responses to your comments are listed below.
> The paper analyzes a technique for the Hessian update in a quadratic function model using function values only. The analysis extends some previous results to the case where the update is constructed from just a few function values. I think the paper is quite interesting and generally well written and, therefore, I recommend publication in its present form. Minor typos: f is not defined at line 94.
We fixed the minor typo you found.